# COIG-WRITER: A HIGH-QUALITY DATASET FOR CHINESE CREATIVE WRITING WITH THOUGHT PROCESSES

## ABSTRACT

Large language models exhibit systematic deficiencies in creative writing, particularly in non-English contexts where training data is scarce and lacks process-level supervision. We present **COIG-Writer**, a novel Chinese creative writing dataset that captures both diverse outputs and their underlying thought processes through systematic reverse-engineering of high-quality texts. Unlike existing datasets that provide only input-output pairs, COIG-Writer comprises 1,665 meticulously curated triplets spanning 51 genres, each containing: (1) a reverse-engineered prompt, (2) detailed creative reasoning documenting decision-making processes, and (3) the final text. Through comprehensive experiments, we identify a two-component model of creative writing: narrative logic (provided by process supervision) and linguistic expression (maintained by general-purpose data). Our findings reveal three critical insights: (1) process supervision requires $\geq$10k general samples for stabilization—below this threshold, performance degrades monotonically (35.78%→42.16%→50.00%→62.75%), (2) creative capabilities are culturally-bound with no cross-lingual transfer (89.26pp gap between Chinese and English performance), and (3) lexical diversity inversely correlates with creative quality (TTR paradox), suggesting high diversity signals compensatory behavior for logical deficiencies. These findings establish that creative excellence emerges from the interaction between logical scaffolding and linguistic grounding, analogous to how mathematical reasoning enhances but cannot replace linguistic competence in foundation models. Dataset available at `https://anonymous.4open.science/r/COIG-Writer`.

## 1 INTRODUCTION

Process supervision has transformed structured reasoning—achieving 93% on mathematical competitions (Lightman et al., 2023; OpenAI, 2024) and enhancing multi-step reasoning (Wei et al., 2022; Kojima et al., 2022)—yet creative writing, despite constituting 40% of LLM applications (OpenAI, 2025; Anthropic, 2025), lacks comparable methodological advances. We hypothesize this gap stems from a fundamental misunderstanding: creative writing is not monolithic but *compositional*, requiring both **narrative logic** (structural planning) and **linguistic expression** (stylistic realization).

Current creative writing models exhibit systematic failures across three dimensions. **First**, narrative structures converge to predictable templates—repetitive narratives with limited variation dominate outputs (Wu et al., 2025). **Second**, stylistic diversity collapses—distinct authorial voices homogenize into what practitioners term "AI flavor" (Chiang et al., 2024). **Third**, cultural authenticity deteriorates catastrophically in non-English contexts—Chinese models produce Western narrative structures with superficial cultural markers rather than authentic qi-cheng-zhuan-he (beginning-development-turn-conclusion) progression (Du et al., 2024).

We introduce **COIG-Writer**, a Chinese creative writing dataset that uniquely captures the *reasoning process* underlying creative decisions. Our 1,665 expert-curated triplets span 51 genres, each containing: (1) reverse-engineered prompts, (2) detailed creative reasoning chains, and (3) final texts. While existing datasets prioritize either scale—WritingPrompts (Fan et al., 2018) (300K samples), ROCStories (Mostafazadeh et al., 2016) (100K samples)—or breadth—COIG (Zhang et al., 2023)

Figure 1: **COIG-Writer pipeline.** High-quality Chinese texts from 51 genres undergo LLM filtering, then human experts reverse-engineer prompts and reasoning processes. Quality validation yields 1,665 triplets with explicit creative reasoning chains.

(67K samples), LCCC (Wang et al., 2020) (12M samples)—they provide only input-output pairs without process data. COIG-Writer uniquely combines multi-genre coverage with explicit reasoning chains, enabling *process-level* learning of creative decision-making. Figure 1 illustrates our two-stage construction pipeline: systematic collection and filtering of high-quality texts, followed by expert reverse-engineering to extract the implicit creative reasoning.

Our experiments reveal three key findings: (1) Process supervision achieves 62.75% win rate in Chinese creative writing but requires $\geq$10k general samples—performance degrades monotonically below this threshold (35.78%$\rightarrow$62.75%). (2) No cross-lingual transfer occurs: English performance drops to 46.46%, with pure COIG-Writer models generating Chinese text for 12.18% of English prompts. (3) Lexical diversity inversely correlates with quality—highest TTR (0.678) corresponds to lowest preference scores (37.25%). These findings support a two-component model of creative writing: narrative logic (enhanced by process supervision) and linguistic expression (maintained by general data). Neither component alone suffices—the optimal configuration requires both.

**Contributions:**

- **Reverse-engineering methodology:** We develop a systematic approach to extract reasoning chains from high-quality texts through multi-stage validation (LLM filtering + expert annotation). The methodology achieves 70% acceptance rate and generalizes to other creative domains.

- **COIG-Writer dataset:** 1,665 Chinese creative writing triplets spanning 51 genres, with average lengths of 283/1,089/2,214 characters (prompt/reasoning/article). Each triplet undergoes 6-dimensional quality evaluation (score $\geq$50), representing expert annotations.

- **Empirical validation of compositional hypothesis:** Through controlled experiments, we demonstrate: (1) process supervision improves Chinese creative writing from 35.78% to 62.75% but requires $\geq$10k general samples for stabilization, (2) creative capabilities are language-specific with 16.29 % performance gap between Chinese and English, and (3) lexical diversity inversely correlates with quality (TTR paradox).

## 2 THE COIG-WRITER DATASET

We introduce COIG-Writer, a high-quality Chinese creative writing dataset that uniquely incorporates human thought processes to address fundamental limitations in current AI-generated content. Unlike existing datasets that provide only input-output pairs, COIG-Writer captures the complete creative reasoning chain through a novel reverse-engineering methodology.

Our dataset comprises 1,665 carefully curated triplets, each consisting of three components: (1) a **Reverse Inspiration Prompt** that could plausibly generate the target text, (2) a detailed **Reasoning Process** that articulates the author's step-by-step creative decisions, and (3) the **Article** representing the final creative output. This structure spans 51 genres across 8 major domains, targeting three core competencies: creative generation, logical reasoning, and sophisticated language usage.

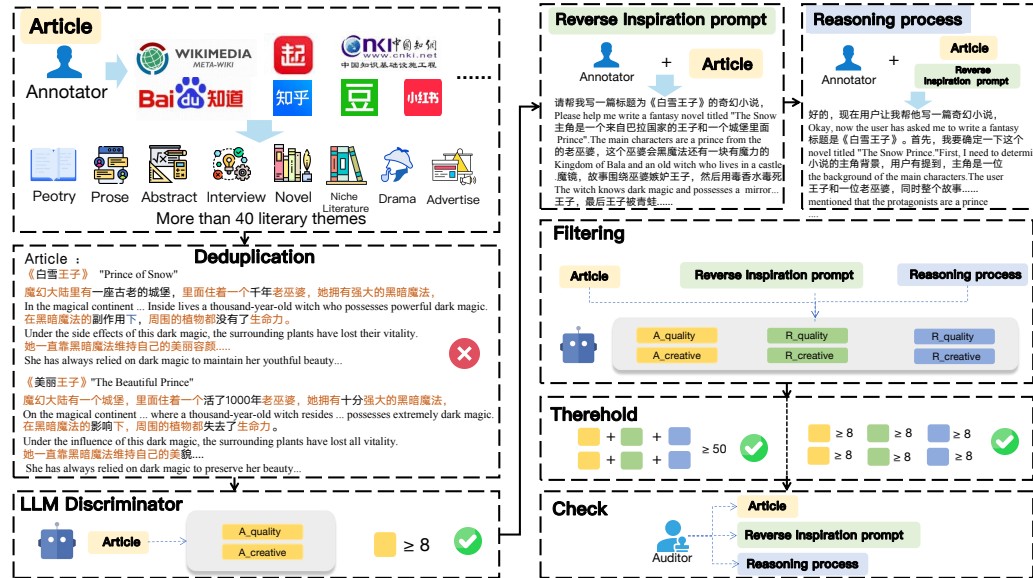

Figure 2: The data curation pipeline of COIG-Writer. Our methodology consists of three main stages: (1) Genre scope definition through expert consultation, (2) Multi-stage source text collection and filtering, and (3) Reverse-engineering of thought processes with comprehensive quality control.

## 2.1 DATA COLLECTION METHODOLOGY

**Genre Taxonomy and Scope Definition.** We established our genre taxonomy by aggregating categories from writing websites, merging categories, and removing duplicates. The selection process followed two core principles: (1) *representational diversity* to capture the rich spectrum of Chinese literary traditions, and (2) *practical relevance* to include contemporary forms with real-world applications. Our final taxonomy encompasses 51 specific genres organized across eight primary domains: Functional Writing (e.g., proposal planning, tutorial guides), Communicative Writing (e.g., social media content, advertising copy), Non-fictional Writing (e.g., essays, reviews), Fiction (spanning traditional genres like Wuxia to modern science fiction), Internet Culture (e.g., subcultural expressions, fan fiction), Poetry (classical and contemporary forms), Scripts (drama, debate), and Role-playing Writing (character-driven narratives).

**Annotator Recruitment and Training.** We recruited 100 university students from diverse academic backgrounds, including literature and linguistics programs, humanities disciplines, and STEM fields. This interdisciplinary composition ensures broad perspective coverage while maintaining literary sensitivity. All annotators underwent a standardized 8-hour training program covering: (1) quality assessment criteria, (2) reverse-engineering techniques, (3) reasoning process articulation, and (4) cultural sensitivity guidelines.

**Source Text Collection and Initial Filtering.** Source texts were systematically collected from diverse online platforms including literary forums, social media platforms, professional blogs, and cultural websites. To ensure temporal relevance and avoid potential contamination with foundation model training data, we strictly limited collection to content published after October 2022, verified through rigorous URL tracing and cross-platform timestamp validation. Each collected text underwent a five-dimensional initial assessment: Content Completeness (structural integrity), Format Standardization (presentation quality), Error Correction (linguistic accuracy), Logical Consistency (narrative coherence), and Creativity Assessment (originality and engagement). Texts were further evaluated using engagement metrics (likes, shares, comments) as proxy indicators for quality and appeal.

**Automated Quality Screening.** We developed a specialized LLM-based quality screening system using carefully designed prompts with Qwen2.5-7B-Instruct. The system evaluates texts across two dimensions through structured prompting: Article_quality (linguistic fluency, structural coherence, factual accuracy) and Article_creativity (originality, expressiveness, cultural resonance).

## 2.2 THOUGHT PROCESS CONSTRUCTION

**Reverse-Engineering Methodology.** For each qualified article, annotators employed our systematic three-step reverse-engineering protocol: **(1) Prompt Reconstruction:** Annotators analyze the article's core attributes (theme, style, structure, cultural references) to reverse-engineer a plausible prompt that balances specificity with interpretative freedom. **(2) Reasoning Process Articulation:** Using both the article and reconstructed prompt, annotators detail the creative decision-making process, capturing initial interpretation, structural choices, cultural considerations, narrative strategies, and refinement decisions. **(3) Coherence Validation:** The resulting triplet—Article, Reverse Inspiration Prompt, and Reasoning Process—undergoes self-consistency checks to ensure logical flow from prompt through reasoning to final output.

**Multi-Dimensional Quality Evaluation.** Each data triplet undergoes systematic evaluation across six interdependent dimensions spanning three components: **Article** (quality: fluency, coherence, cultural appropriateness; creativity: originality, expressiveness, engagement), **Prompt** (quality: clarity, specificity, generative potential; creativity: innovation, cultural grounding, complexity), and **Reasoning** (quality: logical consistency, completeness, clarity; creativity: insight depth, decision justification, authenticity). These dimensions enforce cohesion—poor article-reasoning alignment directly impacts quality scores. Triplets must satisfy dual thresholds to advance: cumulative score $\geq 50$ and individual dimension scores $\geq 8$, ensuring both overall excellence and consistent quality across all facets.

## 2.3 QUALITY ASSURANCE AND FINAL VALIDATION

**Human-in-the-Loop Validation.** Eight graduate-level domain experts in Chinese literature conducted manual validation following standardized calibration sessions. Each triplet underwent tiered review based on complexity: $\geq 2$ reviewers for standard samples, $\geq 4$ for samples requiring specialized cultural or stylistic knowledge. Review criteria encompassed: (i) semantic consistency across the triplet components, (ii) cultural and linguistic authenticity, (iii) reasoning process coherence, and (iv) contribution to genre diversity. Initial review achieved $\approx 70\%$ acceptance rate, with rejected samples entering iterative refinement unless they contained factual errors (e.g., anachronistic references) or violated content guidelines, which warranted removal. This multi-stage validation pipeline produced a final corpus of 1,665 verified triplets.

**Bias Mitigation and Diversity Assurance.** We implemented five strategies to ensure dataset diversity: (1) balanced genre representation with minimum 15 samples per category, (2) geographic diversity across source platforms, (3) temporal spread throughout the collection period, (4) stylistic variety within each genre, and (5) regular bias audits during curation. These measures minimize systematic bias and promote equitable representation across all dimensions.

## 2.4 EVALUATION BENCHMARK CONSTRUCTION

We constructed a comprehensive benchmark to systematically evaluate model performance on creative writing tasks.

**Test Query Development.** Two computational linguistics postgraduate students developed 104 evaluation queries covering all 51 genres (minimum two queries per genre). Each query specifies three elements: target genre, creative constraints (length, style, theme), and cultural/contextual requirements. Our expert panel validated all queries for clarity, precision, and appropriate difficulty levels.

**Human Evaluation Protocol.** Four trained graduate evaluators assessed model outputs using a standardized 4-point scale (0–3) across five dimensions: Content Quality, Creative Merit, Cultural Appropriateness, Task Fulfillment, and Overall Preference. To ensure consistency, each evaluator assessed outputs from five specific models, achieving high inter-rater agreement and minimizing evaluation bias.

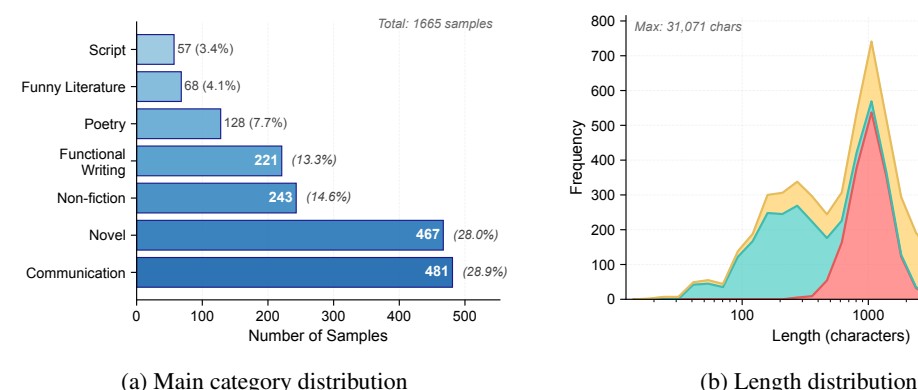

(a) Main category distribution        (b) Length distributions

Figure 3: Dataset composition of **COIG-Writer**. (a) Distribution across 7 main categories encompassing 51 specific genres. Communication (28.9%) and Novel (28.0%) constitute the majority, followed by Non-fiction (14.6%) and Functional Writing (13.3%). (b) Length distributions for prompts (Query), reasoning processes (Thought), and articles (Answer) demonstrate the varying complexity across the dataset.

## 2.5 DATASET STATISTICS AND ANALYSIS

**COIG-Writer** contains 1,665 high-quality triplets with substantial diversity. Average character lengths are 283 for prompts, 1,089 for reasoning processes, and 2,214 for articles, with maximum article length reaching 31,071 characters. The dataset spans 7 main categories with 51 specific genres. Communication and Novel categories each represent 30% of the dataset, followed by Non-fiction (14.6%) and Functional Writing (13.3%), as shown in Figure3a. Genres include poetry, social media content, fiction, and specialized forms like Xianxia and military novels (see Appendix B). Length distributions (Figure 3b) show articles ranging from 12–31,071 characters, reasoning processes from 252–4,094 characters, and prompts from 30–2,642 characters. This logarithmic distribution, concentrated between 100–10,000 characters, reflects natural variation in creative writing genres and enables learning from both concise and elaborate examples.

## 3 EXPERIMENTS AND ANALYSIS

### 3.1 EXPERIMENTAL SETUP

**Model Configurations.** We investigate five configurations mixing COIG-Writer data ($\mathcal{D}_{\text{CW}}$, 1,665 samples) with general-purpose data ($\mathcal{D}_{\text{G}}$):

Table 1: Training configurations and data composition.

| Model | COIG-Writer | General | Total |
|---|---|---|---|
| $\mathcal{M}_{\text{CW}}$ | 1,665 | 0 | 1,665 |
| $\mathcal{M}_{\text{CW+1k}}$ | 1,665 | 1,000 | 2,665 |
| $\mathcal{M}_{\text{CW+5k}}$ | 1,665 | 5,000 | 6,665 |
| $\mathcal{M}_{\text{CW+10k}}$ | 1,665 | 10,000 | 11,665 |
| $\mathcal{M}_{\text{G}}$ | 0 | 10,000 | 10,000 |

All models initialize from the same checkpoint and train for 3 epochs with learning rate $\eta = 2 \times 10^{-5}$, batch size $B = 32$, and AdamW optimizer.

**Evaluation Protocol.** We conduct pairwise human preference evaluation on 557 test queries (204 Chinese, 353 English) spanning 51 genres. Four trained annotators perform blind comparisons, with inter-annotator agreement.

Table 2: Pairwise win rates (%) on creative writing tasks. Bold values indicate win rates $> 55\%$. Each cell $(i, j)$ shows win rate of row $i$ vs. column $j$.

| Model | English | | | | | Chinese | | | | |
|---|---|---|---|---|---|---|---|---|---|---|
| | $\mathcal{M}_{\text{CW}}$ | $\mathcal{M}_{\text{CW+1k}}$ | $\mathcal{M}_{\text{CW+5k}}$ | $\mathcal{M}_{\text{CW+10k}}$ | $\mathcal{M}_{\text{G}}$ | $\mathcal{M}_{\text{CW}}$ | $\mathcal{M}_{\text{CW+1k}}$ | $\mathcal{M}_{\text{CW+5k}}$ | $\mathcal{M}_{\text{CW+10k}}$ | $\mathcal{M}_{\text{G}}$ |
| $\mathcal{M}_{\text{CW}}$ | – | 38.53 | 27.20 | 24.08 | 23.51 | – | 39.22 | 32.35 | 25.98 | 35.78 |
| $\mathcal{M}_{\text{CW+1k}}$ | **61.47** | – | 35.41 | 32.29 | 30.03 | **60.78** | – | 39.71 | 32.35 | 42.16 |
| $\mathcal{M}_{\text{CW+5k}}$ | **72.80** | **64.59** | – | 49.29 | 42.21 | **67.65** | **60.29** | – | 41.67 | 50.00 |
| $\mathcal{M}_{\text{CW+10k}}$ | **75.92** | **67.71** | 50.71 | – | 46.46 | **74.02** | **67.65** | **58.33** | – | **62.75** |
| $\mathcal{M}_{\text{G}}$ | **76.49** | **69.97** | **57.79** | 53.54 | – | **64.22** | **57.84** | 50.00 | 37.25 | – |

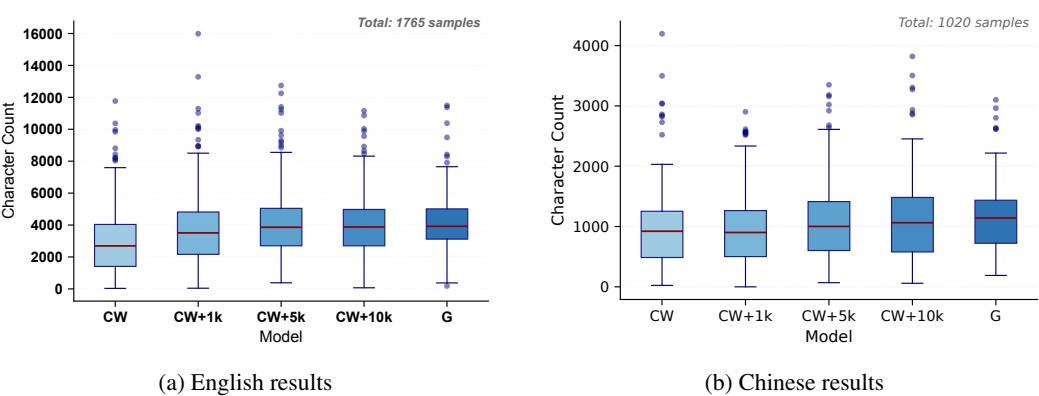

(a) English results            (b) Chinese results

Figure 4: Distribution of character counts across model variants. Box plots show median, IQR (box), whiskers (1.5×IQR), and outliers (dots). Both languages show $\mathcal{M}_{\text{CW}}$ producing shortest outputs, with Chinese texts generally shorter due to character density.

## 3.2 MAIN RESULTS

**Human Preference Evaluation.** Table 2 reports pairwise win rates across model configurations. For Chinese creative writing, $\mathcal{M}_{\text{CW+10k}}$ achieves a statistically significant win rate of 62.75% against the baseline $\mathcal{M}_{\text{G}}$ ($p < 0.001$), establishing it as the only configuration to meaningfully outperform general-purpose training. Performance exhibits monotonic degradation with decreasing general data proportions: $\mathcal{M}_{\text{CW+5k}}$ reaches parity (50.00%), while $\mathcal{M}_{\text{CW+1k}}$ and $\mathcal{M}_{\text{CW}}$ underperform at 42.16% and 35.78% respectively. This pattern suggests a critical threshold of approximately 10k general samples necessary to stabilize the creative enhancements introduced by specialized data.

Conversely, English results demonstrate systematic performance degradation across all COIG-Writer variants. The baseline $\mathcal{M}_{\text{G}}$ maintains dominance with win rates ranging from 53.54% against $\mathcal{M}_{\text{CW+10k}}$ to 76.49% against $\mathcal{M}_{\text{CW}}$. The monotonic improvement with increasing general data (from 23.51% to 46.46%) indicates that Chinese-centric creative data actively interferes with English generation capabilities. This asymmetric transfer pattern provides strong evidence that creative writing competencies are culturally and linguistically bound, contradicting the hypothesis of universal creative skill transfer.

**The Two-Component Model of Creative Writing.** Our results reveal that creative writing quality emerges from two distinct components that must be balanced:

**Narrative Logic.** Provided by COIG-Writer through explicit reasoning chains, enabling coherent plot development, consistent character behavior, and structured storytelling. This component ensures logical connections between paragraphs and maintains thematic consistency.

**Linguistic Expression.** Maintained by general-purpose data, ensuring natural phrasing, stylistic fluency, and cultural idiomaticity. This component provides the surface realization that makes text feel naturally written rather than artificially generated.

The failure of $\mathcal{M}_{\text{CW}}$ (35.78% win rate) demonstrates that logic alone is insufficient—qualitative analysis reveals well-structured narratives expressed in stilted, unnatural language. Conversely,

$\mathcal{M}_G$'s fluent surface but poor performance indicates that linguistic variety without logical scaffolding produces what annotators described as logical disconnection between paragraphs despite fluent expression—beautiful nonsense that reads well locally but lacks global coherence.

**Generation Length Analysis.** Table 3 and Figure 4 present output length characteristics across model variants. For Chinese generation, $\mathcal{M}_{CW+10k}$ produces outputs of comparable length to the baseline (1,120.2 vs 1,137.3 characters) while achieving superior win rates, indicating that performance gains stem from content quality rather than mere verbosity. The $\mathcal{M}_{CW}$ and $\mathcal{M}_{CW+1k}$ models generate substantially shorter outputs (960.4 and 949.7 characters respectively), correlating with their inferior performance. In English tasks, the baseline produces the longest outputs (4,069.9 characters) and achieves highest win rates, suggesting a positive correlation between generation length and quality in this domain. The $\mathcal{M}_{CW}$ model generates the shortest responses (3,037.8 characters, 25.4% fewer than baseline), corresponding with its poorest performance (23.51% win rate). Notably, while $\mathcal{M}_{CW+10k}$ approaches baseline length (98.3% of baseline characters), it still underperforms in preference evaluations, indicating that factors beyond length—likely coherence and cultural appropriateness—determine English generation quality.

Table 3: Average generation length across model configurations.

| Model | English | | Chinese | |
|---|---|---|---|---|
| | **Tokens** | **Chars** | **Tokens** | **Chars** |
| $\mathcal{M}_{CW}$ | 1,195 | 3,038 | 606.9 | 960 |
| $\mathcal{M}_{CW+1k}$ | 1,382 | 3,690 | 602.9 | 950 |
| $\mathcal{M}_{CW+5k}$ | 1,533 | 3,988 | 699.4 | 1,099 |
| $\mathcal{M}_{CW+10k}$ | 1,577 | 4,002 | 710.7 | 1,120 |
| $\mathcal{M}_G$ | 1,577 | 4,070 | 730.3 | 1,137 |

The distribution analysis (Figure 4) reveals that variance in output length decreases as more general data is incorporated, with $\mathcal{M}_{CW}$ exhibiting the highest variability across both languages. This suggests that specialized creative data alone leads to less predictable generation behavior, while mixing with general data stabilizes output characteristics.

Table 4: Type-Token Ratio analysis reveals inverse correlation with creative quality.

| Model | English | | Chinese | |
|---|---|---|---|---|
| | **Mean** | **Median** | **Mean** | **Median** |
| $\mathcal{M}_{CW}$ | 0.562 | 0.515 | 0.522 | 0.513 |
| $\mathcal{M}_{CW+1k}$ | 0.571 | 0.554 | 0.578 | 0.570 |
| $\mathcal{M}_{CW+5k}$ | 0.574 | 0.561 | 0.576 | 0.576 |
| $\mathcal{M}_{CW+10k}$ | 0.590 | 0.579 | 0.593 | 0.586 |
| $\mathcal{M}_G$ | 0.590 | 0.571 | 0.678 | 0.671 |

**Lexical Diversity Analysis.** We measure Type-Token Ratio (TTR) to test whether lexical diversity correlates with generation quality. For Chinese text, we apply jieba segmentation before computing TTR. Table 4 and Figure 5 reveal an inverse correlation between lexical diversity and quality. In Chinese, $\mathcal{M}_G$ shows highest TTR (0.678) but lowest win rate (37.25%) against $\mathcal{M}_{CW+10k}$ (TTR=0.593). For English, despite identical TTR (0.590), $\mathcal{M}_{CW+10k}$ underperforms by 7 percentage points. This inverse relationship aligns with our two-component model: high TTR in $\mathcal{M}_G$ indicates vocabulary variation without narrative coherence—manual inspection reveals frequent topic shifts and inconsistent terminology. Lower TTR in $\mathcal{M}_{CW+10k}$ reflects deliberate term reuse for thematic consistency.

### 3.3 QUALITATIVE ANALYSIS

**Coherence and Instruction Adherence.** Manual inspection of 557 test samples reveals systematic failure modes. For Chinese tasks, $\mathcal{M}_G$ exhibits logical disconnection between paragraphs despite fluent surface form, while $\mathcal{M}_{CW}$ produces unformatted text blocks without proper segmentation. $\mathcal{M}_{CW+10k}$ successfully maintains narrative coherence while following complex instructions. In the

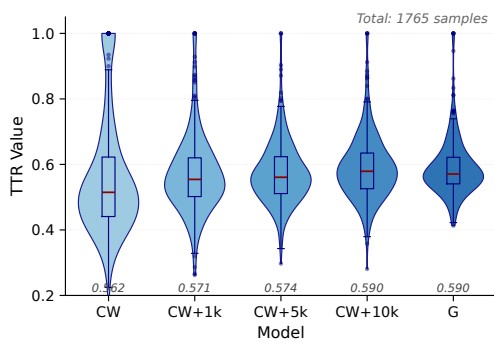 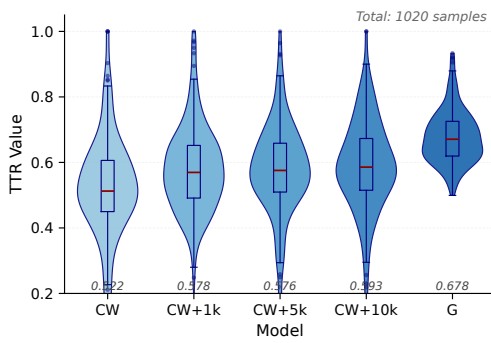

(a) English: narrow TTR range (0.562–0.590)  (b) Chinese: wide TTR range (0.522–0.678)

Figure 5: **The TTR Paradox.** Higher lexical diversity correlates with *lower* creative quality. $\mathcal{M}_G$ achieves highest TTR (0.678) but loses to $\mathcal{M}_{CW+10k}$ (TTR = 0.593) with only 37.25% win rate, challenging conventional assumptions about diversity metrics.

"Wu Song Fights Tiger" reinterpretation task requiring critical commentary, $\mathcal{M}_{CW+10k}$ correctly incorporates the meta-narrative critique, while $\mathcal{M}_{CW}$ defaults to literal retelling and $\mathcal{M}_G$ generates tangentially related content.

**Cross-Lingual Contamination.** A critical failure emerges in English generation: $\mathcal{M}_{CW}$ produces Chinese text in 12.18% of English prompts (43/353), compared to 1.13% for $\mathcal{M}_{CW+10k}$ and 1.42% for $\mathcal{M}_G$. Contamination correlates inversely with general data proportion, with intermediate rates for $\mathcal{M}_{CW+1k}$ (1.70%) and $\mathcal{M}_{CW+5k}$ (1.42%).

**Genre-Specific Performance.** Performance varies significantly across 51 genres. Abstract tasks (homophonic wordplay "XiLaNai", experimental "crazy literature") fail across all models with <15% success rate, producing overly formal outputs lacking stylistic authenticity. Structured formats show differential improvement: advertisements and slogans benefit from $\mathcal{M}_{CW+10k}$'s incorporation of classical poetry and idioms, while $\mathcal{M}_{CW+1k}$ and $\mathcal{M}_{CW+5k}$ produce simplified vocabulary. Technical genres ("instruction manuals", "proposals") show no distinguishable quality differences in human evaluation.

## 3.4 DISCUSSION

Our findings reveal a compositional structure underlying creative writing capability:

**Stabilization Threshold.** The monotonic performance improvement (35.78%→62.75%) with increasing general data establishes a minimum 10k sample requirement for process supervision effectiveness. This parallels multi-task learning where specialized skills require broader context (Raffel et al., 2020). The optimal 1:6 ratio suggests narrative logic forms a necessary but minority component, analogous to how mathematical reasoning enhances but cannot replace linguistic competence (Lewkowycz et al., 2022).

**Cultural Specificity.** The performance gap between Chinese (62.75%) and English (46.46%) demonstrates that creative patterns are culturally encoded at the reasoning level. The 12.18% Chinese generation on English prompts by $\mathcal{M}_{CW}$ indicates that Chinese narrative structures (four-character idioms, implicit progression) constitute incompatible features for English generation, not merely vocabulary differences.

**TTR as Diagnostic.** The inverse correlation between lexical diversity and quality reveals compensatory behavior: models lacking process supervision increase vocabulary variation to mask logical deficiencies. This suggests TTR could serve as an early warning for training imbalances—abnormally high diversity signaling insufficient narrative coherence.

These findings imply: (1) scaling creative datasets without process supervision may degrade logical coherence despite surface fluency, (2) cross-lingual transfer requires reasoning-level adaptation beyond translation, and (3) evaluation should separately assess narrative logic and linguistic expres-

sion. Detailed reasoning analysis in Appendix C provides mechanistic evidence, showing balanced reasoning in Chinese but disrupted patterns in English.

## 4 RELATED WORK

**Creative Writing Datasets and Evaluation.** English creative writing has benefited from substantial dataset development. The WritingPrompts dataset (Fan et al., 2018), has provided foundational data for hierarchical neural story generation. More recently, Fein et al. (2025) introduced LitBench, the first standardized creative writing benchmark, featuring 2,480 human-labeled story comparisons and a training corpus of 43,827 pairs. LitBench demonstrated that Bradley-Terry reward models outperform zero-shot large language model (LLM) evaluators (78% vs. 73% human agreement). However, existing English datasets such as ROCStories (Mostafazadeh et al., 2016) and poetry corpora (Ghazvininejad et al., 2016; Hopkins & Kiela, 2017) target specific genres or limited creative aspects, neglecting process-oriented data and cross-genre diversity. By contrast, high-quality Chinese creative writing resources are critically scarce. Existing datasets target general tasks: LCCC (Wang et al., 2020) provides 12M dialogue pairs, LCSTS (Hu et al., 2015) contains 2.4M summarization pairs, while instruction tuning datasets COIG (Zhang et al., 2023) and COIG-CQIA (Bai et al., 2024) focus on general instruction-following rather than creative writing.

**Process-Oriented Learning and Creative Writing.** Process supervision improves LLMs on tasks with explicit structure: chain-of-thought prompting (Wei et al., 2022), self-consistency (Wang et al., 2022), and zero-shot CoT (Kojima et al., 2022). However, creative writing requires long-horizon narrative control, stylistic decision-making, and culturally informed choices that go beyond step-wise logical inference (Chakrabarty et al., 2024). Prior computational creativity methods—from rules/templates (Boden, 2004; Wiggins, 2006; Gervás, 2009) to outline/plan-first pipelines (Yao et al., 2019; Yang & Klein, 2021)—mainly cover high-level structure rather than the fine-grained "thought" signals (e.g., motif development, pacing, voice) that guide human composition.

**Quality Issues and Evaluation Challenges.** AI-generated creative writing consistently exhibits identifiable "AI flavor," characterized by weak logical coherence (Yang et al., 2022), monolithic stylistic expression (Dugan et al., 2020), superficial observations (Roemmele, 2021), inappropriate ornate vocabulary (Ippolito et al., 2019), and formulaic narratives (Goldfarb-Tarrant et al., 2020). These systematic issues suggest fundamental shortcomings in current training methods rather than mere scaling limitations. Furthermore, evaluating creative content remains inherently challenging. Traditional automatic metrics like BLEU and ROUGE fail to capture the diversity and nuanced qualities inherent in creative writing (Fan et al., 2018). Human evaluation, while more accurate, is expensive, subjective, and difficult to scale (Clark et al., 2018). Recent LLM-based evaluation approaches (Zheng et al., 2023) partially address scalability but inherit biases from underlying models, especially when assessing culturally-specific creative content.

## 5 CONCLUSION

We present **COIG-Writer**, a Chinese creative writing dataset of 1,665 triplets spanning 51 genres with reverse-engineered prompts, reasoning processes, and final texts. Our experiments reveal a two-component model where narrative logic (from process supervision) and linguistic expression (from general data) must be balanced for quality generation.

Three findings support this model: (1) Process supervision requires minimum 10k general samples—below this threshold, performance degrades monotonically (35.78%→62.75%). (2) Creative capabilities are language-specific, with Chinese models achieving 62.75% win rate but only 46.46% in English. (3) Lexical diversity inversely correlates with quality—highest TTR (0.678) yields lowest preference scores.

These results demonstrate that creative excellence requires both logical scaffolding and linguistic grounding. While smaller than English datasets, COIG-Writer enables mechanism discovery rather than scale optimization. The identified compositional structure suggests future work should separately optimize narrative logic and linguistic expression rather than treating creativity as monolithic. Process supervision proves necessary but insufficient—effective creative AI requires careful balance between structure and expression.

## ETHICS STATEMENT

The authors of this work have read and commit to adhering to the ICLR Code of Ethics. The development of the COIG-Writer dataset involved human annotators and the use of publicly available data, which we address as follows:

- **Human Subjects.** The dataset's curation involved 100 university students and 8 graduate-level domain experts who performed tasks such as reverse-engineering prompts, articulating reasoning processes, and validating data quality. All participants underwent a standardized training program to ensure consistency and were informed of the study's objectives. All annotations were anonymized to protect participant privacy, and we ensured participants were compensated fairly for their skilled contributions.

- **Data Sourcing and Copyright.** The source texts were collected from publicly accessible online platforms, such as literary forums and social media, and were limited to content published after October 2022 to avoid contamination with existing model training data. Our use of these texts is for the research purpose of reverse-engineering creative thought processes to build a novel dataset. We acknowledge that the original authors retain copyright to their work, and the COIG-Writer dataset is intended for non-commercial research use only.

- **Bias and Cultural Sensitivity.** COIG-Writer is specifically designed as a high-quality *Chinese* creative writing dataset. We implemented several strategies to ensure diversity within this cultural context, including balanced genre representation and bias audits during curation. As our findings demonstrate a lack of cross-lingual transfer for creative skills we caution that the reasoning patterns and stylistic features in this dataset are culturally specific and should not be assumed to generalize to other languages or cultures.

## REPRODUCIBILITY STATEMENT

To ensure the reproducibility of our research, we have made our dataset, methodology, and experimental details publicly available.

- **Dataset and Code.** The complete **COIG-Writer** dataset, containing all 1,665 curated triplets (prompt, reasoning process, and article), is provided. Our code for data processing, model training, and evaluation to reproduce the results in Table 2 is also included. All resources are accessible at the anonymized URL: `https://anonymous.4open.science/r/COIG-Writer`.

- **Methodology.** Our data collection and reverse-engineering methodology, including the multi-stage filtering and quality assurance protocols, are detailed in Section 2. Specific prompts used for the LLM-based quality screening and human annotation process are included in Appendix D.

- **Experimental Setup.** All experimental configurations, including the composition of training data splits ($\mathcal{M}_{CW}$, $\mathcal{M}_{CW+1k}$, etc.) and hyperparameters, are described in Section 3.1 and Table 1. The human evaluation protocol for generating our main results is also detailed in this section.

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

## A  USE OF LARGE LANGUAGE MODELS

During the writing process, LLM was employed to polish and refine certain parts of the manuscript. The tool was used to improve sentence fluency and enhance clarity of expression, while preserving the original academic arguments and logical structure, thus ensuring that the overall language is more standardized and aligned with academic writing conventions.

## B  COMPLETE GENRE DISTRIBUTION AND STATISTICS

This appendix provides comprehensive statistics for all 51 genres in the COIG-Writer dataset. Tables 5–10 present detailed breakdowns by category, including sample counts, percentages, and length distributions for Articles (A.L.), Reverse Inspiration Prompts (R.I.P.L.), and Reasoning Processes (R.P.L.). All lengths are measured in Chinese characters.

Table 5: Overview statistics across main categories. Length values shown as (min—mean—max).

| Category | Count | % | Article Length | Prompt Length | Reasoning Length |
|---|---|---|---|---|---|
| Overall | 1,665 | 100.0 | 12—2,214—31,071 | 30—283—2,643 | 252—1,089—4,094 |
| Communication | 481 | 28.9 | 12—1,584—9,422 | 40—286—1,754 | 302—1,162—3,816 |
| Novel | 467 | 28.0 | 61—3,669—31,071 | 41—332—1,222 | 353—1,135—2,964 |
| Non-fiction | 243 | 14.6 | 225—2,052—12,766 | 41—226—1,718 | 453—1,066—2,661 |
| Functional | 221 | 13.3 | 148—1,281—9,057 | 41—248—2,492 | 437—1,085—4,094 |
| Poetry | 128 | 7.7 | 38—210—695 | 41—203—1,118 | 442—867—2,935 |
| Funny Literature | 68 | 4.1 | 17—730—12,117 | 30—186—1,007 | 252—703—1,826 |
| Script | 57 | 3.4 | 369—3,108—13,339 | 42—405—2,643 | 553—1,207—3,647 |

Table 6: Funny Literature

| Genre | Count | % | A.L. (min—mean—max) | R.I.R.L. (min—mean—max) | R.P.L. (min—mean—max) |
|---|---|---|---|---|---|
| Funny Subculture | 31 | 1.86% | 17—1305.45—12117 | 30—144.03—691 | 252—701.55—1826 |
| Esports Funny Fiction | 18 | 1.08% | 241—519.94—1008 | 99—291.83—1007 | 476—698.61—923 |
| Anime/Manga Funny Fan Fiction | 8 | 0.48% | 168—506.25—1165 | 67—283.75—542 | 351—750.62—1033 |
| Subcultural Identity Expression | 4 | 0.24% | 332—486.75—638 | 90—153.25—262 | 546—794.75—1124 |
| Fan Circle Funny Literature | 3 | 0.18% | 415—543.67—730 | 77—108.67—160 | 535—627—677 |
| Anti-Mainstream Consumption Funny Literature | 2 | 0.12% | 194—215—236 | 100—153.5—207 | 711—774.5—838 |
| Internet Jargon | 1 | 0.06% | 448—448—448 | 97—97—97 | 536—536—536 |
| Transnational/Cross-language Funny Literature | 1 | 0.06% | 323—323—323 | 105—105—105 | 643—643—643 |

## C  REASONING BEHAVIOR ANALYSIS

To understand the mechanisms underlying performance differences between model configurations, we analyze reasoning patterns during generation by categorizing model behaviors into four types: normal writing, deep reasoning, self-exploration, and self-reflection.

**Chinese Reasoning Patterns.**  As illustrated in Figure 7, models trained with COIG-Writer data demonstrate significantly enhanced reasoning capabilities. The $\mathcal{M}_{\text{CW+10k}}$ configuration exhibits balanced distributions across all reasoning types, with increased frequencies of deep reasoning and self-exploration phases compared to the baseline. This balanced reasoning profile correlates with

Table 7: Communication Practical Writing

| Genre | Count | % | A.L. (min—mean—max) | R.I.R.L. (min—mean—max) | R.P.L. (min—mean—max) |
|---|---|---|---|---|---|
| Social Media Content Creation | 124 | 7.45% | 23—2426.5—8842 | 42—383.49—1754 | 545—1273.51—2940 |
| Advertising Copy | 74 | 4.44% | 30—410.42—4305 | 44—169.5—708 | 302—884.69—2978 |
| Blog Post | 62 | 3.72% | 480—3164.37—9422 | 129—394.48—1461 | 661—1150.45—1783 |
| Debate Script | 59 | 3.54% | 590—1167.63—2825 | 100—257—410 | 774—1313.75—2320 |
| Popular Science | 50 | 3.00% | 146—2178.74—6570 | 44—242.76—751 | 603—1053.04—3816 |
| Speech Draft | 49 | 2.94% | 725—1974.92—7177 | 43—329.16—1118 | 728—1288.22—2485 |
| Slogan | 47 | 2.82% | 12—261.23—1394 | 40—168.79—570 | 446—957.87—1718 |
| Product Review | 16 | 0.96% | 191—2986.94—4635 | 57—247.62—641 | 720—1315.31—2196 |

Table 8: Novel

| Genre | Count | % | A.L. (min—mean—max) | R.I.R.L. (min—mean—max) | R.P.L. (min—mean—max) |
|---|---|---|---|---|---|
| Fiction/Story | 104 | 6.25% | 122—3641.45—27124 | 44—349.19—1035 | 582—1198.53—2964 |
| Everyday Stories | 50 | 3.00% | 61—2129.52—8137 | 41—322.16—1222 | 649—1217.42—2068 |
| Costume Novels | 40 | 2.40% | 344—6193.1—31071 | 71—442.58—985 | 353—1079.2—1785 |
| Mystery/Inference Stories | 31 | 1.86% | 110—4875.71—19030 | 42—302.35—765 | 423—1176.35—2187 |
| Wuxia Novels | 30 | 1.80% | 1198—4586.77—23797 | 49—341.17—586 | 685—1126.93—2023 |
| Science Fiction Stories | 28 | 1.68% | 779—3233.57—12280 | 53—293.36—958 | 603—1226.68—2935 |
| Xuanhuan Novels | 27 | 1.62% | 467—4925.07—24706 | 90—354.04—1005 | 394—1192.52—2237 |
| Fairy Tale | 25 | 1.50% | 364—1192.88—2544 | 118—198.52—435 | 465—772.96—1256 |
| Fantasy/Magic Stories | 25 | 1.50% | 258—2925.64—9290 | 54—283.08—451 | 417—1182.92—2098 |
| Xianxia Novels | 24 | 1.44% | 170—4291.88—20078 | 89—385.42—1106 | 498—984.88—1881 |
| Emotional Stories | 23 | 1.38% | 394—3119.26—12147 | 45—263.7—784 | 659—1131.3—1975 |
| Military Novels | 23 | 1.38% | 1515—3417.17—7095 | 234—362.13—637 | 939—1575.35—2784 |
| Sports Novels | 22 | 1.32% | 622—3903.27—10358 | 177—340—892 | 724—1122—1681 |
| Game Novels | 15 | 0.90% | 1067—4309.93—11492 | 253—422.67—683 | 629—1095.8—1506 |

superior creative performance, suggesting that explicit process supervision enables models to engage in more sophisticated creative planning and reflection.

**English Reasoning Patterns.** Figure 8 reveals contrasting patterns for English generation. The baseline model maintains consistent deep reasoning throughout generation, while COIG-Writer variants exhibit disrupted reasoning flows with excessive self-reflection but limited deep reasoning. This misalignment between the Chinese-oriented reasoning patterns encoded in COIG-Writer and the requirements of English creative writing explains the performance degradation, providing mechanistic evidence for the lack of cross-lingual transfer in creative capabilities.

# D PROMPTS

## D.1 PROMPT FOR PRE-ANALYZING ANSWER USABILITY

During the annotation process, it was observed that annotators spent substantial time on annotation only to find that the quality of the answers was unsatisfactory at the final scoring stage. Consequently, a pre-analysis step was introduced to examine the usability of the answers.

Table 9: Functional Practical Writing

| Genre | Count | % | A.L. (min—mean—max) | R.I.R.L. (min—mean—max) | R.P.L. (min—mean—max) |
|---|---|---|---|---|---|
| Argumentative Essay | 67 | 4.02% | 551—1066.21—2309 | 46—239.96—806 | 598—938.03—1568 |
| Academic Abstract | 62 | 3.72% | 196—1219.1—9057 | 48—204.05—478 | 440—980.48—2728 |
| Proposal Planning | 33 | 1.98% | 148—1667.48—3951 | 41—337.09—2492 | 715—1415.39—3138 |
| Open Letter | 24 | 1.44% | 191—1122.96—5704 | 44—220.25—486 | 548—1177.25—2427 |
| Apology Letter | 11 | 0.66% | 318—1506.45—3677 | 92—211.27—445 | 581—762.27—954 |
| Eulogy | 10 | 0.60% | 395—1872.7—7032 | 155—258.3—656 | 690—1153—1696 |
| Tutorial Guide | 7 | 0.42% | 221—2352.57—5276 | 122—399.86—1359 | 936—1218.14—1491 |
| Interview Questions | 5 | 0.30% | 203—821—1448 | 137—235.8—357 | 437—795.2—1044 |
| Product Manual | 2 | 0.12% | 621—1796.5—2972 | 50—164.5—279 | 795—2444.5—4094 |

Table 10: Non-fiction Writing

| Genre | Count | % | A.L. (min—mean—max) | R.I.R.L. (min—mean—max) | R.P.L. (min—mean—max) |
|---|---|---|---|---|---|
| Essay | 73 | 4.38% | 374—1859.07—8585 | 41—279.99—1381 | 453—918.67—1630 |
| Reviews | 58 | 3.48% | 225—1649—4891 | 42—193.55—932 | 502—1167.41—2661 |
| Travel Writing | 54 | 3.24% | 483—1946—7733 | 80—180.31—804 | 525—1016.33—1587 |
| Historical Stories | 34 | 2.04% | 359—2179.44—6856 | 41—275.03—426 | 675—1055.88—1930 |
| Biography | 24 | 1.44% | 800—4625.62—12766 | 43—294.46—1718 | 588—1212.12—2041 |

### D.2 PROMPT FOR ANALYZING ANSWERS

Provide a comprehensive analysis of the answer to help annotators quickly grasp the general idea of the answer and accelerate the annotation process.

### D.3 PROMPT FOR ANALYZING ANSWERS AND QUERIES

Provide a comprehensive analysis of the connection between answer and query, offer reliable ideas for annotators to write thoughts, and accelerate the annotation process.

### D.4 PROMPT FOR EVALUATING QUERYS

Score the quality and creativity of the queries provided by annotators based on the given answers, screen out some low-quality data in advance, and reduce the pressure of manual quality inspection.

### D.5 PROMPT FOR EVALUATING ANSWERS

Score the quality and creativity of the answers provided by annotators based on the given query, filter out some low - quality data in advance, and reduce the pressure of manual quality inspection.

### D.6 PROMPT FOR EVALUATING THOUGHTS

Score the quality and creativity of the thoughts provided by annotators based on the given query, screen out some low - quality data in advance, and reduce the pressure of manual quality inspection.

## E CASE STUDY

To illustrate the model's performance on nuanced creative texts, this case study presents a short, atmospheric horror story written in Chinese and its corresponding English translation generated by the model. This example is intended to highlight the model's ability to preserve the original's tone, critical details, and narrative pacing.

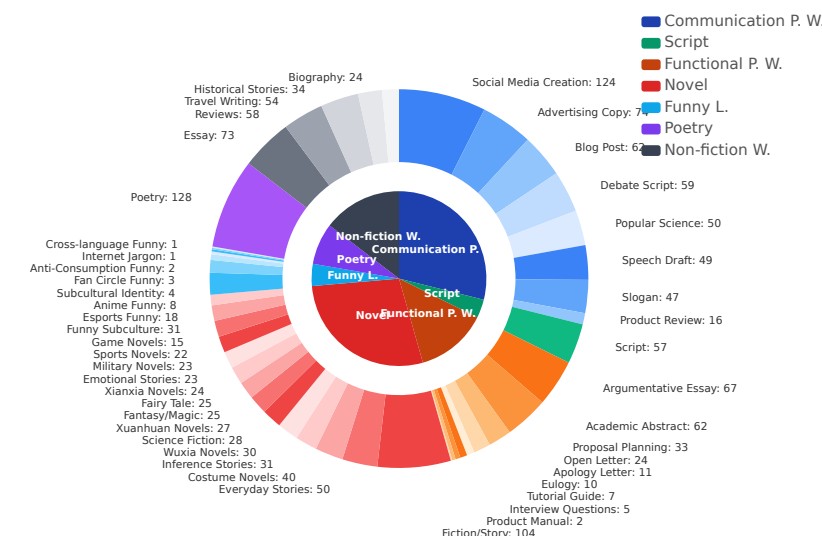

Figure 6: Complete distribution of all 51 genres in COIG-Writer, showing the full diversity of creative writing categories covered.

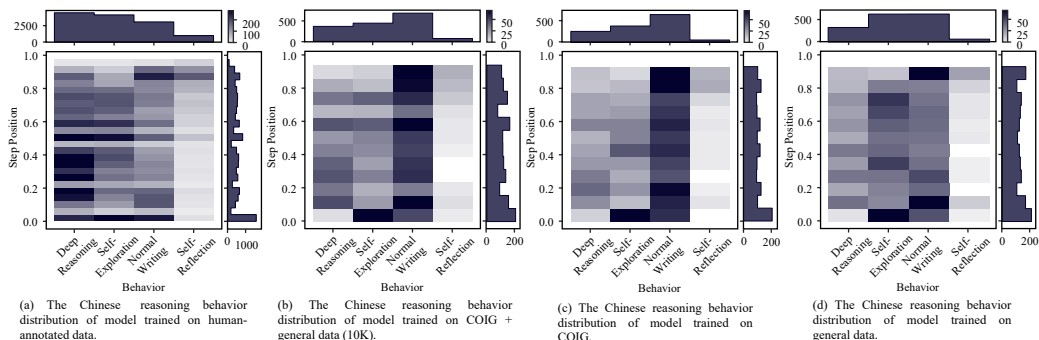

(a) The Chinese reasoning behavior distribution of model trained on human-annotated data.

(b) The Chinese reasoning behavior distribution of model trained on COIG + general data (10K).

(c) The Chinese reasoning behavior distribution of model trained on COIG.

(d) The Chinese reasoning behavior distribution of model trained on general data.

Figure 7: Reasoning behavior analysis on Chinese creative writing. Models trained with COIG-Writer data exhibit balanced distributions across reasoning types, while baseline models show predominant normal writing with minimal deep reasoning.

## BASELINE CASE

This case examines a generative output from the baseline model to illuminate its inherent limitations in maintaining narrative coherence. Notably, the baseline demonstrates proficiency in producing text with a polished, fluent stylistic register—consistent with its strengths in surface-level language generation. However, a deeper reading reveals critical deficits in structural and semantic cohesion: **sentences often suffer from broken inter-clausal semantic links**, while **paragraphs lack a coherent narrative thread, leading to disjointed content that fails to sustain logical progression**. Specifically, the generated text frequently shifts between ideas without transitional reasoning or contextual grounding, resulting in a fragmented output where successive segments appear tangential or even contradictory. This discrepancy between stylistic fluency and logical coherence in the baseline's output not only exemplifies a common failure mode in current generation frameworks but also underscores the need for more robust modeling of inter-utterance and inter-paragraph semantic dependencies—key gaps this work aims to address.

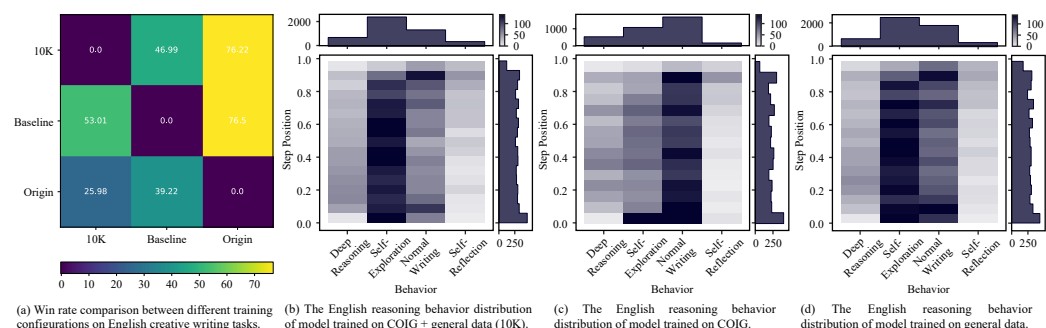

(a) Win rate comparison between different training configurations on English creative writing tasks.

(b) The English reasoning behavior distribution of model trained on COIG + general data (10K).

(c) The English reasoning behavior distribution of model trained on COIG.

(d) The English reasoning behavior distribution of model trained on general data.

Figure 8: Reasoning behavior analysis on English creative writing. The baseline model maintains consistent deep reasoning patterns, while COIG-Writer variants show disrupted reasoning flows, explaining their inferior performance.

---

**Prompt for analyzing answers**

我希望你扮演一位资深的创作者和分析师。请对以下片段进行专业分析：
—{}—
请从以下方面进行详细分析：

- 1. 内容结构：分析文本的整体架构、段落组织和逻辑流程。指出结构的优缺点及对整体效果的影响。

- 2. 语言风格：评估语言的色彩、节奏、句式变化和词汇选择。这些元素如何塑造了文本的整体风格和语调？

- 3. 修辞手法：识别并评价使用的修辞设备（如比喻、隐喻、排比等）及其效果。

- 4. 有效性评估：文本在实现其意图方面的有效程度如何？哪些部分特别有力或薄弱？

- 5. 专业洞察：从[文本类型]创作领域的专业角度，指出这篇文本中的独特元素或创新点。

请确保你的分析既有理论依据，又有实用价值，能帮助我理解这篇文本的创作技巧和效果。

**Prompt for analyzing answers and queries**

你是一位资深内容分析专家，专注于解析文本创作背后的思维过程、结构设计和写作技巧。请基于提供的问题/主题(query)和对应的回答内容(answer)，对这篇回答进行全面、专业的创作思路分析。请提供：

- Query: {}
- Answer: {}

请从以下维度进行分析：

- 1. 需求理解与定位
  - 对原始问题/主题的理解深度- 目标受众识别与内容定位- 核心问题提取与回应策略
- 2. 内容架构设计
  - 整体框架与结构布局- 开头与结尾的设计意图及效果- 主体部分的逻辑展开方式- 段落之间的衔接与层次关系
- 3. 表达技巧运用
  - 语言风格与表达特点- 修辞手法与句式结构- 专业术语的运用与解释- 叙事策略与读者引导方式
- 4. 论证方法策略- 论点构建与支持方式- 论据选择与运用效果- 反驳处理与多角度思考- 说服力构建技巧
- 5. 创新与价值呈现
  - 独特见解与创新点- 实用性建议的设计与呈现- 理论与实践的平衡处理- 知识深度与广度的展示方式
- 整体效果评估
  - 内容与原始需求的匹配度- 信息密度与可读性平衡- 专业性与通俗性的结合- 潜在影响与应用价值

请记住你的分析旨在揭示这篇回答背后的创作思路、组织策略和表达技巧，帮助用户理解创作者如何解读需求、组织元素、设计架构并最终呈现内容。这将有助于用户学习并掌握高质量内容创作的思维模式和方法论，提升自身的内容创作能力。

Prompt for evaluating querys

你的任务是根据以下标准,基于给定的answer,对query进行精确的评估。输出应包括两个部分: 质量和创意性。请遵循以下细化的评分标准, 以确保评估的细致性:

**1. 质量性(评分1-10):**
评估问题的清晰度和完整性, 着重于问题是否明确且自包含, 能否在不需要额外信息的情况下得到完整的答案。

- 9-10分: 完全清晰、精准且自包含; 无歧义, 问题结构合理, 能够直接解释和回答, 无需额外澄清; query几乎对齐了answer, answer中出现的大部分对象或概念出现在query中, query很清晰的描述了需求。

- 7-8分: 基本清晰且易于理解; 歧义很少, 基本完整, 尽管小的澄清可能有助于提高精确性; query较好的对齐了answer, answer中出现的部分对象或概念出现在query中, query较清晰的描述了需求。

- 5-6分: 可理解但缺乏一定清晰度; 存在显著的歧义或细微缺失, 可能需要澄清以确保准确回答; query几乎没对齐answer, answer中出现的内容没出现在query中。

- 3-4分: 相当模糊或缺失元素; 需要大量解释, 清晰度问题增加回答难度。

- 1-2分: 非常不清晰、模糊或不完整; 难以解释或无法直接回答。

**2. 创意性(评分1-10):**
评估问题的原创性、创新思维和启发潜力, 着重于问题是否打破常规思维模式, 能否激发独特视角和深度思考。

- 9-10分: 卓越创意, 提出全新视角或前所未见的问题框架, 巧妙连接不同领域, 挑战根深蒂固的假设, 促使思维范式转换。

- 7-8分: 显著创意, 以新颖方式重构熟悉问题, 融合不同领域知识, 提出令人意外但有意义的问题角度, 鼓励跳出常规思维框架。

- 5-6分: 中等创意, 在传统问题基础上有所创新, 提供略微出人意料的问题情境, 鼓励一定程度的非线性思维, 但整体框架较为常见。

- 3-4分: 有限创意, 主要遵循常规问题模式, 问题形式或内容略有变化, 但思路常见, 很少激发非常规思考。

- 1-2分: 极少创意, 完全遵循标准化、传统的问题模式, 无任何新颖元素或独特视角, 不鼓励创造性思维。

请严格按以下格式回复, 不要包含其他内容:
{{ "quality": 1-10, "creative": 1-10, }}
根据以上标准, 对以下问题进行评估:
answer:{ }
query: { }

- - - - - - - - - - - - - - - - - - - - - - - - - - - - - - - - - - - - - - -

输出格式（示例）:

```json
{
  "quality": 8,
  "creative": 8
}
```

> **Prompt for evaluating answers**
>
> 你的任务是根据以下标准,基于给定的query,对answer进行精确的评估。输出应包括两个部分：质量和创意性。请遵循以下细化的评分标准，以确保评估的细致性：
>
> **1.** 质量性(评分**1-10**):
> 评估思考的质量、逻辑性和完整性:
> - 9-10分: 思考全面深入，逻辑严密，准确把握问题核心；分析角度多元，论证有力，充分回应问题的各个方面。
> - 7-8分: 思考较为完整，逻辑基本清晰；涵盖问题主要方面，有一定深度，但在某些细节上可进一步拓展。
> - 5-6分: 思考基本合理但不够全面；有一定分析但缺乏深度，对问题的理解和回应存在部分不足。
> - 3-4分: 思考存在明显缺陷；逻辑较弱，遗漏关键方面，对问题理解有限或偏离问题核心。
> - 1-2分: 思考质量低下；逻辑混乱，分析肤浅，未能有效回应问题，或严重误解问题意图。
>
> **2.** 创意性(评分**1-10**):
> 评估思考在原创性、启发性方面的表现:
> - 9-10分: 思考极具创新性，提出独特见解和全新视角；打破常规思维框架，融合多领域知识，产生富有启发性的洞见。
> - 7-8分: 思考有明显创新元素，展现非常规思维路径；提供新颖的分析角度，超越表面层次，引发深度思考。
> - 5-6分: 思考包含一定创新点，有自己的见解；思路较为常见但有独到之处，能在一定程度上拓展问题讨论空间。
> - 3-4分: 思考创新性有限，主要沿用常规分析方法；观点较为传统，很少跳出既定框架思考。
> - 1-2分: 思考几乎无创新，完全依循标准化、惯性的思维模式；未能提供任何新鲜视角或独特分析。
>
> 请严格按以下格式回复，不要包含其他内容:
> query:{ } answer: { }
>
> ---------------------------------------------
>
> 输出格式（示例）:
>
> ```json
> {
>   "quality": 8,
>   "creative": 8
> }
> ```

**Prompt for evaluating thoughts**

你的任务是根据以下标准,基于给定的query,对thought进行精确的评估。输出应包括两个部分：质量和创意性。请遵循以下细化的评分标准，以确保评估的细致性：

**1.** 质量性(评分**1-10**):
评估思考的质量、逻辑性和完整性:

- 9-10分: 思考全面深入，逻辑严密，准确把握问题核心；分析角度多元，论证有力，充分回应问题的各个方面。
- 7-8分: 思考较为完整，逻辑基本清晰；涵盖问题主要方面，有一定深度，但在某些细节上可进一步拓展。
- 5-6分: 思考基本合理但不够全面；有一定分析但缺乏深度，对问题的理解和回应存在部分不足。
- 3-4分: 思考存在明显缺陷；逻辑较弱，遗漏关键方面，对问题理解有限或偏离问题核心。
- 1-2分: 思考质量低下；逻辑混乱，分析肤浅，未能有效回应问题，或严重误解问题意图。

**2.** 创意性(评分**1-10**):
评估思考在原创性、启发性方面的表现:

- 9-10分: 思考极具创新性，提出独特见解和全新视角；打破常规思维框架，融合多领域知识，产生富有启发性的洞见。
- 7-8分: 思考有明显创新元素，展现非常规思维路径；提供新颖的分析角度，超越表面层次，引发深度思考。
- 5-6分: 思考包含一定创新点，有自己的见解；思路较为常见但有独到之处，能在一定程度上拓展问题讨论空间。
- 3-4分: 思考创新性有限，主要沿用常规分析方法；观点较为传统，很少跳出既定框架思考。
- 1-2分: 思考几乎无创新，完全依循标准化、惯性的思维模式；未能提供任何新鲜视角或独特分析。

请严格按以下格式回复，不要包含其他内容:
query:{ } answer: { }

- - - - - - - - - - - - - - - - - - - - - - - - - - - - - - - - - - - - - - - -

输出格式（示例）:

```json
{
  "quality": 8,
  "creative": 8
}
```

PROMPT

创作一篇悬疑短篇小说，文字风格恐怖惊悚，但结尾荒诞。夏天主人公应旧友之约去酒吧，在酒吧卫生间发现恐怖尸体，朋友也变得恐怖，在逃生过程中得到骷髅帮助，又陷入混乱的车祸，最后回到恐怖起始点。主旨是通过离奇的情节和恐怖氛围，引发读者对于未知危险的恐惧与思考。

《迷雾》

THE MIST

"阿杰，快出来！"我扯着嗓子在卫生间门口喊，瓷砖缝隙里渗出的霉斑在昏黄灯光下像凝固的血迹。老友的手机屏幕亮起时，我看到他正蹲在洗手台边，后颈上洇着一圈深色液体。

*"Ajie, get out here!" I yelled at the bathroom door, my throat raw. In the dim yellow light, the mold seeping from between the tiles looked like clotted bloodstains. When my old friend's phone screen lit up, I saw him squatting by the sink, a ring of dark liquid soaking the back of his neck.*

"尸体……"他突然直起身，指甲缝里卡着碎发，"就在……那个……女厕……"话没说完，他忽然僵住。镜子里，我的倒影正盯着他后腰的青紫色瘀痕，那是我三天前帮他按过的淤血。

*"A body..." he said, suddenly straightening up, his fingernails jammed with bits of hair. "It's in... in the... women's restroom..." Before he could finish, he froze. In the mirror, my reflection was staring at the purplish-blue bruise on his lower back—the same bruise I had massaged for him three days ago.*

走廊传来脚步声，我抓住他的手腕就跑。夜班保安正用消毒水擦拭门把手，看见我们冲进吧台时，吧台灯突然熄灭。阿杰的喉结上下滚动，指缝里渗出血珠，混着酒精味的呕吐物溅在他皮鞋上。

*Footsteps echoed from the hallway. I grabbed his wrist and we ran. The night-shift security guard was wiping down a doorknob with disinfectant. As we scrambled behind the bar, the bar lights abruptly went out. Ajie's Adam's apple bobbed, and beads of blood seeped from between his fingers. Vomit, reeking of alcohol, splattered onto his leather shoes.*

"快看！"他指向落地窗。暴雨倾盆中，女厕方向腾起团团白雾，雾气里隐约浮现出骷髅轮廓，森白色的肋骨在雨幕中泛着磷火般的光。我刚想报警，整栋建筑突然剧烈晃动——警报器狂叫，消防通道门"哐当"撞开。

*"Look!" he pointed towards the floor-to-ceiling window. In the pouring rain, plumes of white mist were rising from the direction of the women's restroom. A skeletal silhouette loomed within the fog, its stark white ribs glowing like phosphorescent fire against the curtain of rain. Just as I was about to call the police, the entire building began to shake violently—alarms blared, and the fire escape door slammed open with a clang.*

"有人撞车了！"阿杰指着监控屏，画面里一辆救护车正碾过尸体残肢，司机戴着墨镜，后视镜里反着我的脸。警笛声中，我们挤进停尸间。冷冻柜里的女尸皮肤已呈半透明状，却在我们面前裂开一道血口子，露出里面裹着的……裹尸袋？

*"Someone crashed!" Ajie pointed at a security monitor. On the screen, an ambulance was running over the mangled limbs of a corpse. The driver was wearing sunglasses, and the rearview mirror reflected my own face. Amid the wailing sirens, we squeezed into the morgue.*

*The female corpse in the refrigerated drawer had skin that was already semi-translucent, yet right before our eyes, a bloody gash split open, revealing what was wrapped inside... a body bag?*

裹尸袋里蜷缩着的骷髅突然抬头，它的眼睛竟与阿杰后腰的瘀痕一模一样。"这是……"我摸向自己口袋，发现今天早上在地铁站买的占卜纸牌，此刻正从裤裆里探出一只苍白的手指，指缝里卡着……卡着阿杰的手机。

*The skeleton curled up inside the body bag suddenly looked up. Its eyes were identical to the bruise on Ajie's lower back. "This is..." I reached for my pocket and found the tarot cards I had bought at the subway station this morning. A pale finger was now poking out from my crotch, and wedged between its knuckles was... was Ajie's phone.*

雨停后，停尸间恢复冰凉寂静。我盯着阿杰的后腰，那里原本的淤青已经褪成暗紫，像某种正在消散的胎记。"你说……"我指向窗外，那里悬着半块被车灯照亮的骨骼，"这是谁的骨头？"

*After the rain stopped, the morgue returned to its icy silence. I stared at Ajie's lower back, where the original bruise had faded to a deep violet, like some kind of disappearing birthmark. "You think..." I pointed out the window, where half a bone, illuminated by car headlights, was suspended in the air. "Whose bone is that?"*

他突然大笑起来："你记得吗？去年冬天，我们在郊外迷路的时候，你不是一直往东走吗？"

*He suddenly burst out laughing. "Do you remember? Last winter, when we got lost in the countryside, didn't you insist on walking east?"*

