# OpenReview forum: "COIG-Writer: A High-Quality Chinese Creative Writing with Thought Process Dataset"
_ICLR.cc/2026/Conference — ICLR 2026 Conference Withdrawn Submission_

### Official Review · Reviewer_1Dz9 · 2025-10-31

**Soundness:** 1
**Presentation:** 2
**Contribution:** 1
**Rating:** 2
**Confidence:** 4

**Summary:**

This paper introduces COIG-Writer, a new dataset for Chinese creative writing. The authors propose a "reverse-engineering" methodology to create a dataset of 1,665 triplets, each containing a prompt, a high-quality article, and a synthetic "reasoning process" chain intended to capture the creative decisions. The dataset spans 51 genres.
Based on experiments with this dataset, the authors make several claims:
1. A "two-component model" of creative writing exists, composed of Narrative Logic (from process supervision) and Linguistic Expression (from general data).
2. Process supervision is only effective when "stabilized" by a threshold of general-purpose data (e.g., $\ge$10k samples in this setup), with performance degrading monotonically below.
3. Creative capabilities are culturally-bound, with the paper finding no cross-lingual transfer from its Chinese-centric dataset to English tasks.
4. A "TTR Paradox" exists, where higher lexical diversity (Type-Token Ratio) inversely correlates with creative quality.

**Strengths:**

1. The paper correctly identifies a lack of high-quality, process-oriented datasets for non-English creative writing, specifically Chinese.
2. The goal of capturing creative reasoning is ambitious. The dataset's structure (prompt-reasoning-article) is novel.
3. The inclusion of 51 distinct genres across 8 domains is a strength, providing broad stylistic coverage despite the small sample size.
4. While the explanation is flawed, the observation of an inverse correlation between TTR and human preference is an interesting data point for the community.

**Weaknesses:**

1. The paper's premise rests on "reverse-engineered thought processes". This is a synthetic, post-hoc rationalization. The paper provides no evidence that these chains are valid, reliable, or representative of actual human creative cognition. No Inter-Annotator Agreement (IAA) for the reasoning generation task is provided. This is a fatal methodological flaw.
2. The paper must include a baseline trained on the 1,665 articles without the 1,089-character reasoning chains. Without this ablation, it is impossible to know if the "process supervision" (the reasoning) adds any value, or if the performance gains simply come from fine-tuning on 1,665 high-quality, in-domain articles.
3. The paper's main finding (the "10k stabilization threshold") is confounded. The best model is trained on 11,665 total samples, while the baseline $\mathcal{M}_{G}$ is trained on 10,000. The results cannot distinguish between the effect of "process supervision" and the effect of simply having 16.65% more training data. The monotonic improvement is also just as likely to be a standard result of mitigating catastrophic forgetting by data mixing, not evidence of a "two-component model."
4. The dataset (N=1,665) is too small to be a significant contribution. The paper's own experiments show models trained on it alone ($\mathcal{M}_{CW}$) are non-functional, severely limiting its utility.
5. The paper fails to compare its models against any SOTA baselines (e.g., zero-shot performance of strong foundation models like GPT-5, Gemini 2.5pro, or Claude sonnet 4.5) or on established creative writing benchmarks (e.g., LitBench, WritingBench).
6. The human evaluation uses only four trained evaluators, with annotators assigned to specific models, introducing a risk of rater-model confounds. No IAA is reported for the preference evaluation.

**Questions:**

See the weaknesses above

---

### Official Review · Reviewer_6RES · 2025-11-01

**Soundness:** 2
**Presentation:** 3
**Contribution:** 2
**Rating:** 2
**Confidence:** 3

**Summary:**

The paper presents COIG-Writer, a novel dataset for Chinese creative writing that includes not only creative outputs but also reverse-engineered prompts and detailed reasoning chains. The dataset spans 1,665 samples across 51 genres, aiming to address deficiencies in large language models (LLMs) for creative writing in non-English contexts. The authors conduct experiments to analyze the impact of process supervision and general data on model performance, proposing a two-component model of creative writing: narrative logic and linguistic expression.

**Strengths:**

Novelty: First dataset to systematically capture creative reasoning processes in Chinese writing, filling a major gap in non-English resources.
Genre Diversity: Covers 51 genres and 7 major categories, ensuring broad representational coverage.
Rigorous Curation: Multi-stage filtering, expert annotation, and human-in-the-loop validation for high data quality.
Empirical Insights: Experiments highlight the need to balance narrative logic and linguistic expression for creative excellence in LLMs.
Cultural Sensitivity: The work demonstrates the lack of cross-lingual transfer in creative writing, emphasizing the importance of culturally specific reasoning.
Open Resource: Dataset and code are made available for reproducibility.

**Weaknesses:**

Insufficient Data Analysis: The paper focuses heavily on model degradation experiments, but lacks a deep, systematic analysis of the dataset itself. For a dataset paper, richer descriptive statistics, qualitative examples, and annotation quality analysis are expected.
Unclear Definition of Creative Writing: There is no rigorous definition of “creative writing” or clear demonstration of how the dataset meets this requirement. The mapping between dataset structure and creative writing criteria is not established.
Ambiguity in Experimental Attribution: It is not entirely clear whether the observed model performance improvements are due to the unique features of COIG-Writer (reasoning chains, genre diversity) or simply increased exposure to more data. The paper does not control for confounding factors or provide ablation studies to isolate the contribution of process supervision.
Evaluation Metrics: Reliance on human preference as the primary evaluation metric may limit reproducibility. Automatic metrics are discussed but not deeply integrated.
Potential Annotator Bias: While bias mitigation strategies are described, the reliance on university students and graduate experts may introduce subtle biases.
Limited Cross-Lingual Analysis: The paper demonstrates poor cross-lingual transfer but does not deeply analyze mechanisms or propose solutions.

**Questions:**

See above

---

### Official Review · Reviewer_BQKJ · 2025-11-01

**Soundness:** 3
**Presentation:** 3
**Contribution:** 2
**Rating:** 4
**Confidence:** 3

**Summary:**

The paper presents 1,665 curated triplets in 51 genres, Each triplet includes: (1) a reverseengineered prompt, (2) detailed creative reasoning documenting decision-making processes, and (3) the final text.

The authors then show that narrative logic and linguistic expression play the role in creative writing.

**Strengths:**

The paper sets very salient question of creative writing and unlike many other papers that try addressing it, provides a rather comprehensive evaluation reporting subjective human evaluations of 100 students.

The observation that there is inverse correlation between linguistic diversity and creativity assessment is thought provoking.

**Weaknesses:**

As always the evaluation is the weakest question around creative writing, though as I mention earlier in this work happens to be above the median for such results, in my opinion, that might be lacking for such a venue as ICLR.

**Questions:**

Some of the texts you list are rather long. How could you guarantee evaluation consistency?

How do you guarantee statistical significance of your win ratios? In particular, could you decompose the magnitudes of several sources of noise in your dataset?

---

### Official Review · Reviewer_pczx · 2025-11-05

**Soundness:** 3
**Presentation:** 3
**Contribution:** 2
**Rating:** 4
**Confidence:** 4

**Summary:**

This paper introduces COIG-Writer, a high-quality dataset for Chinese creative writing that, unlike traditional input-output datasets, includes the underlying thought processes. The authors argue that creative writing is compositional, requiring both "narrative logic" and "linguistic expression," and that existing models fail due to a lack of process-level supervision, especially in non-English contexts. The core contribution is the dataset itself, comprising 1,665 triplets across 51 genres. Each triplet contains: (1) a reverse-engineered prompt, (2) a detailed creative reasoning chain (thought process), and (3) the final high-quality text.

Through experiments training models on this dataset, the paper presents three key findings:

1. **Stabilization Threshold**: Process supervision for creativity is only effective when mixed with a sufficient amount of general-purpose data, with a critical threshold identified around 10,000 general samples.

2. **Cultural Specificity**: Creative writing capabilities are culturally and linguistically bound. Models trained on the Chinese process data failed to improve English writing and exhibited cross-lingual contamination, demonstrating that creative reasoning does not transfer easily.

3. **The "TTR Paradox"**: Lexical diversity (measured by Type-Token Ratio) inversely correlates with human-perceived quality. The highest-TTR models often produced incoherent text, suggesting that high diversity can be a compensatory behavior for a lack of logical structure.

**Strengths:**

1. **Novel Dataset on Creative Writing**: The primary contribution is a high-quality, process-supervised dataset for creative writing, a domain that has been lagging behind more structured tasks like math and coding.

2. **Important Empirical Findings with Counter-intuitive Insights**: The main findings of this paper, especially the two counter-intuitive ones (cultural specificity and TTR paradox) are noteworthy and raises pointers for further investigation.

The "cultural specificity" result is a finding that tempers claims of universal capabilities in LLMs.

The "TTR paradox" is a particularly insightful discovery that challenges a common heuristic in text evaluation, suggesting that high lexical diversity can be a bug, not a feature, if it masks a lack of narrative logic. Traditionally (for humans), lexical diversity is considered positively correlated with creativity, fluency, or sophistication -- especially in second-language acquisition and stylistic analysis (e.g., [3, 4]). This empirically discovered paradox is idiosyncratic to COIG-Writer, and loosely aligns with threads in linguistic literature that stipulates fluency and coherence may rely on controlled lexical reuse rather than maximal variety (e.g., [1, 2]).


### Ref:

1. Hoey, M. (1991). Another perspective on coherence and cohesive. Functional and systemic linguistics: Approaches and uses, 55, 385.

2. Crossley, S., & McNamara, D. (2011). Text coherence and judgments of essay quality: Models of quality and coherence. In Proceedings of the Annual Meeting of the Cognitive Science Society (Vol. 33, No. 33).

3. McKee, Gerard, David Malvern, and Brian Richards. "Measuring vocabulary diversity using dedicated software." Literary and linguistic computing 15.3 (2000): 323-338.

4. McCarthy, Philip M., and Scott Jarvis. "MTLD, vocd-D, and HD-D: A validation study of sophisticated approaches to lexical diversity assessment." Behavior research methods 42.2 (2010): 381-392.

**Weaknesses:**

1. **Limited Scale of the Dataset**: While the quality is extremely high, the dataset size of 1,665 triplets is relatively small for fine-tuning modern LLMs (_largely weakening the 'dataset' contribution of the paper_) , especially when spread across 51 distinct genres. This could limit the generalization capability of models trained exclusively or heavily on this data, particularly for the less-represented genres.

2. **Subjectivity of Reverse-Engineered Reasoning**: The "reasoning process" is a post-hoc rationalization created by an expert annotator, not a direct recording of the original author's true thought process. While the paper's methodology ensures these chains are plausible and coherent, they are fundamentally a simulation. This introduces a potential source of bias, as the dataset may reflect a specific, systematic style of "how to think about writing" rather than a diverse range of actual human creative processes.

3. **Generalizability of the "10k Threshold"**: The 10k general sample threshold for stabilization is a key finding but is likely specific to the experimental setup (base model, size of D_cw, etc.). It is presented as a somewhat firm number, but it is more likely a heuristic that would vary with model size, the quality of the general data, and the size of the process-supervised dataset itself.

**Questions:**

1. The reverse-engineering of thought processes is a fascinating but inherently subjective task. What steps were taken to ensure inter-annotator consistency on the structure and content of the reasoning chains, beyond just ensuring the final triplet was coherent? Is there a risk that this process ingrains a specific, uniform style of "rationalization" into the dataset?

2. The 10k general-sample "stabilization threshold" is a very interesting result. How do you hypothesize this threshold might change with different base models or a larger process-supervised dataset? For instance, if you had 10,000 COIG-Writer samples, would the need for general data decrease, or would the optimal ratio remain similar?

3. **The "TTR paradox" is a powerful finding**. Could you provide a qualitative example from your evaluation that illustrates this? For example, showing a snippet from a baseline MG generation that has high TTR but is logically disconnected, compared to a McW+10k generation with lower TTR but better thematic consistency, would further strengthen this claim.

---

### Note · Authors · 2025-11-17

I have read and agree with the venue's withdrawal policy on behalf of myself and my co-authors.